# A statistical framework for comparing epidemic forests

Cyril Geismar[1,2,3]*, Peter J. White[1,3], Anne Cori[1,3], Thibaut Jombart[1,3]

**1** MRC Centre for Global Infectious Disease Analysis, Imperial College School of Public Health, London, United Kingdom, **2** Bloomberg School of Public Health, Johns Hopkins University, Baltimore, United States of America, **3** NIHR Health Protection Research Unit in Modelling and Health Economics, Imperial College School of Public Health, London, United Kingdom

☯ These authors contributed equally to this work.

\* c.geismar21@imperial.ac.uk

## Abstract

Inferring who infected whom in an outbreak is essential for characterising transmission dynamics and guiding public health interventions. However, this task is challenging due to limited surveillance data and the complexity of immunological and social interactions. Instead of a single definitive transmission tree, epidemiologists often consider multiple plausible trees forming *epidemic forests*. Various inference methods and assumptions can yield different epidemic forests, yet no formal test exists to assess whether these differences are statistically significant. We propose such a framework using a chi-square test and permutational multivariate analysis of variance (PERMANOVA). We assessed each method's ability to distinguish simulated epidemic forests generated under different offspring distributions. While both methods achieved perfect specificity for forests with 100+ trees, PERMANOVA consistently outperformed the chi-square test in sensitivity across all epidemic and forest sizes. Implemented in the R package *mixtree*, we provide the first statistical framework to robustly compare epidemic forests.

## Author summary

Identifying who infected whom is a central part of outbreak investigation. It helps trace the source of infection, uncover missing cases, identify superspreaders, and describe broader dynamics of transmission such as its speed, pattern, and scale. With the advent of pathogen sequencing and digital contact tracing, computational models have become the standard approach for reconstructing outbreaks. These probabilistic models do not identify a single definitive history of who infected whom (*i.e.,* a transmission tree), but a collection of plausible alternatives, which we call *epidemic forests*. Different modelling assumptions or data sources can produce different epidemic forests, but until now, there has been no formal way to determine whether these differences are meaningful. We

**Data availability statement:** Simulations, analyses and visualisations were performed using the R software version 4.4.0 (https://www.R-project.org/) [27]. Our framework has been implemented in a free, open-source R package, mixtree, which is available on CRAN https://cran.r-project.org/web/packages/mixtree/index.html. This study is fully reproducible using code available on GitHub: https://github.com/CyGei/mixtree_analysis (archived on Zenodo: https://doi.org/10.5281/zenodo.17704759) [47]. The resulting data is stored on a Zenodo archive: https://doi.org/10.5281/zenodo.17704456 [48].

**Funding:** This work was supported by the National Institute for Health and Care Research (NIHR) Health Protection Research Unit (HPRU) in Modelling and Health Economics, which was a partnership between Imperial College London, London School of Hygiene & Tropical Medicine, and UKHSA (grant code NIHR200908) (PJW, AC,TJ). CG received a salary from Imperial College London, which was made possible thanks to this funding. The funders had no role in study design, data collection and analysis, decision to publish, or preparation of the manuscript.

**Competing interests:** The authors have declared that no competing interests exist.

present the first statistical framework designed to compare epidemic forests. We evaluate two methods: one that counts how often specific transmission pairs appear, and another that compares the structure of transmission trees. Testing these methods on simulated outbreaks, we found that both successfully identified when forests represented identical transmission dynamics, but one method outperformed the other in identifying forests representing distinct transmission dynamics. Our framework, implemented in the R package mixtree, enables epidemiologists to validate and compare outbreak reconstruction approaches, supporting more reliable investigations.

## Introduction

Tracking who infected whom is central to outbreak investigations. Transmission trees, modelled as directed acyclic graphs (DAGs) where vertices represent infected individuals and directed edges indicate transmission events, delineate infector-infectee relationships [1]. These representations can assist epidemiologists in identifying introduction and superspreading events [2,3], whilst also elucidating broader transmission dynamics relevant to outbreak response. The topology of transmission trees, defined by the arrangement of vertices and edges, encodes key epidemiological parameters. The out-degree distribution of vertices represents the number of secondary infections per infected individual (*i.e.,* the offspring distribution), revealing the extent of heterogeneity in transmission [4–7]. Branching patterns inform on transmission dynamics between groups [8], revealing group reproduction numbers [5] and transmission patterns, for example, between healthcare workers and patients in nosocomial outbreaks [9] or between children and adults in schools [10].

The inference of transmission trees is challenging and often characterised by large uncertainty, partly due to the lack of discriminatory power in choosing between possible transmission pairs [5,9,11], incomplete surveillance data and diverse, sometimes conflicting sources (*e.g.,* contact, temporal, spatial, or genetic data) [12]. Additional complexities arise from varying methodological approaches [12] and pathogen evolution mechanisms that are difficult to model (*e.g.,* within-host evolution and transmission bottleneck [7,13]). Consequently, outbreak reconstruction often yields *epidemic forests*, which are collections of plausible transmission trees rather than a singular definitive representation of who infected whom.

Without formal statistical methods to differentiate epidemic forests, determining whether differences between them represent meaningful variations in transmission dynamics or uncertainty in tree reconstruction is challenging. Such distinction would help validate convergence when repeated model runs produce statistically similar forests and assess whether competing inference approaches or alternative data sources yield significantly different forests.

Bayesian inference methods have emerged as the gold standard for transmission tree reconstruction, with various approaches differing in their assumptions, data requirements, and inference strategies [12]. In this context, epidemic forests

represent samples drawn from a model's posterior distribution of transmission trees. To assess the performance of the inference process, researchers rely on general Markov Chain Monte Carlo (MCMC) diagnostics, applied to scalar parameter chains rather than the inferred trees themselves. These diagnostics evaluate convergence through trace plot inspection and the Gelman-Rubin statistic [14], assess sampling efficiency through effective sample size calculations, and check model fit using posterior predictive checks [15]. In parallel, model selection criteria such as Deviance Information Criterion and Bayesian Information Criterion balance goodness-of-fit with model complexity, enabling comparison of competing inference models fitted to the same data [16]. *Consensus trees* are used also to summarise epidemic forests, typically representing, for each case, the infector with the highest posterior support across samples [11,17–20]. However, these trees are often abstract representations rather than plausible transmission scenarios, potentially introducing cycles or multiple index cases. While algorithms such as Edmonds can enforce a valid tree topology, the resulting consensus tree may correspond to a combination of ancestries that was never observed as a complete tree in the posterior [20–22].

Consequently, standard MCMC diagnostics assess parameter chains rather than the inferred transmission events, model selection criteria compare model fit but not the resulting tree topologies, while consensus trees ignore the uncertainty in who infected whom and may misrepresent key epidemiological features. This underscores the need for specialised statistical methods that can differentiate epidemic forests while accounting for uncertainty and relevant topological properties.

Here, we introduce a statistical framework for testing differences between epidemic forests. We consider two alternative methods: an adaptation of a Monte Carlo chi-square ($\chi^2$) test [23] used to compare the distribution of infector-infectee pairs between forests, and a permutation-based multivariate analysis of variance (PERMANOVA) [24] which compares topological distances between trees within and between forests. Both methods are summarised in Fig 1. We evaluated the performance of each method by comparing simulated epidemic forests with varying offspring distributions, measuring their ability to correctly identify forests stemming from distinct (sensitivity) or identical (specificity) generative processes (see Methods).

## Results

We simulated pairs of epidemic forests that were either stemming from the same, or different generative processes (see Methods). We systematically varied epidemic size ($\epsilon$: 20–200 cases), forest size ($m$: 20–200 trees per forest), and the parameters of the negative binomial offspring distribution ($R_0$: 1.5-3 and $k$: 0.1-Poisson-like), which determine the mean and dispersion of secondary infections. We simulated 90,000 epidemic forests (see Methods) based on which we conducted 5,760,000 tests to measure sensitivity and specificity for the $\chi^2$ test and PERMANOVA.

Overall results are presented as Receiver Operating Characteristic (ROC) curves (supplementary S6 Fig), with area under the curve (AUC) provided in supplementary S7 Fig. Both methods exhibited near-perfect specificity (> 97%), *i.e.,* the ability to correctly identify forests drawn from identical generative processes across all epidemic or forest sizes (Fig 2, row 4). The $\chi^2$ test had a negligible advantage in specificity (+1.5%) when the number of trees in each forest was small ($m \le 50$). Across the aggregated simulation results, sensitivity was near perfect once the forest size reached 50–100 trees, with AUC nearing 1 (supplementary S7 Fig). However, these results varied between methods and simulation settings.

The methods differed substantially in their sensitivity, *i.e.,* their ability to correctly identify forests drawn from different generative processes, with PERMANOVA consistently outperforming the $\chi^2$ test across all scenarios (Fig 2, row 1–3). However, the magnitude of PERMANOVA's advantage varied considerably depending on which parameters differed between forests. A logistic regression model explained 58% of the variance in test sensitivity (pseudo $R^2$ [25] = 0.58), with method choice, forest size, epidemic size, and differences in $R_0$ and $k$ as key predictors (Table 1). Compared to the $\chi^2$ test, PERMANOVA showed much greater sensitivity when forests differed in their dispersion parameter ($\Delta_k$), with 51-fold higher odds of correctly distinguishing overdispersed from Poisson-like forests ($\Delta_{k_{(0,1] \text{ vs. Poisson}}}$), and 8-fold higher odds when comparing forests with different degrees of overdispersion ($\Delta_{k_{(0,1] \text{ vs.} (0,1]}}$) (Table 1). When forests differed in dispersion and

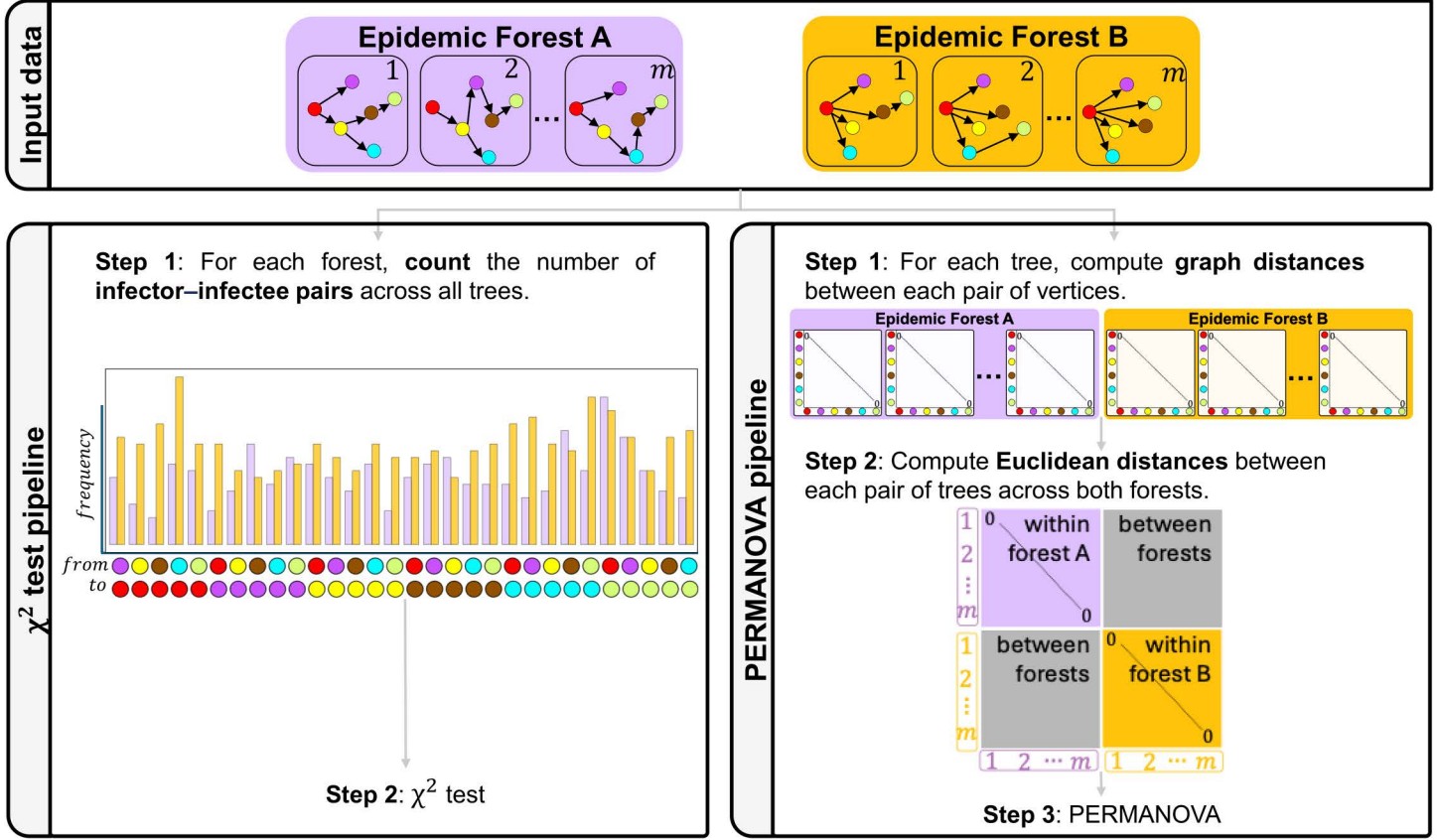

**Fig 1. Statistical framework for comparing epidemic forests.** Diagram illustrating the methods for comparing epidemic forests *A* (pink, *e.g.,* no superspreading) and *B* (orange, *e.g.,* superspreading). Coloured dots represent infected cases, and arrows indicate transmission events. Left: The $\chi^2$ test compares the frequency of infector-infectee pairs between forests. Right: The PERMANOVA method first calculates pairwise standard graph distances (number of transmission events between two cases, plus one – see supplementary S1 Fig) between vertices within each tree, converts these to a Euclidean distance matrix between all trees, and tests for significant topological differences between the forests using permutation-based testing. Both methods test the null hypothesis that the compared epidemic forests stem from the same generative process.

contained at least 100 trees, PERMANOVA achieved near-perfect sensitivity (98.7%), irrespective of epidemic size (Fig 2, rows 1 and 3, column 3).

In contrast, PERMANOVA's advantage over the $\chi^2$ test diminished when forests differed only in reproduction number (OR = 3, Table 1). When both forests shared strong overdispersion (common $k \leq 0.5$), high stochastic variability in individual transmission limited the ability to detect differences in $R_0$ up to 1 ($\Delta R_0 \leq 1$), yielding low sensitivity even with 200 trees per forest (52% across epidemic sizes; supplementary S3 Fig and S4 Fig). Sensitivity improved progressively as the common dispersion parameter approached Poisson-like transmission ($k \to \infty$) or as epidemic size increased (supplementary S4 Fig).

In addition to higher sensitivity, PERMANOVA produced consistently narrower p-value distributions than the $\chi^2$ test, with interquartile ranges substantially smaller across all scenarios (supplementary S2 Fig).

Both methods' sensitivity increased with forest size (OR = 3, 6, 12 for $m = 50$, 100 and 200 respectively) but showed opposite correlations with epidemic size: PERMANOVA's sensitivity rose with larger epidemics (OR= 2, 4 and 5 for $\epsilon$ = 50, 100, and 200 respectively), whereas the $\chi^2$ test's sensitivity declined (OR= 0.5, 0.3, and 0.1) (Table 1, Fig 2). Our findings

PLOS Computational Biology

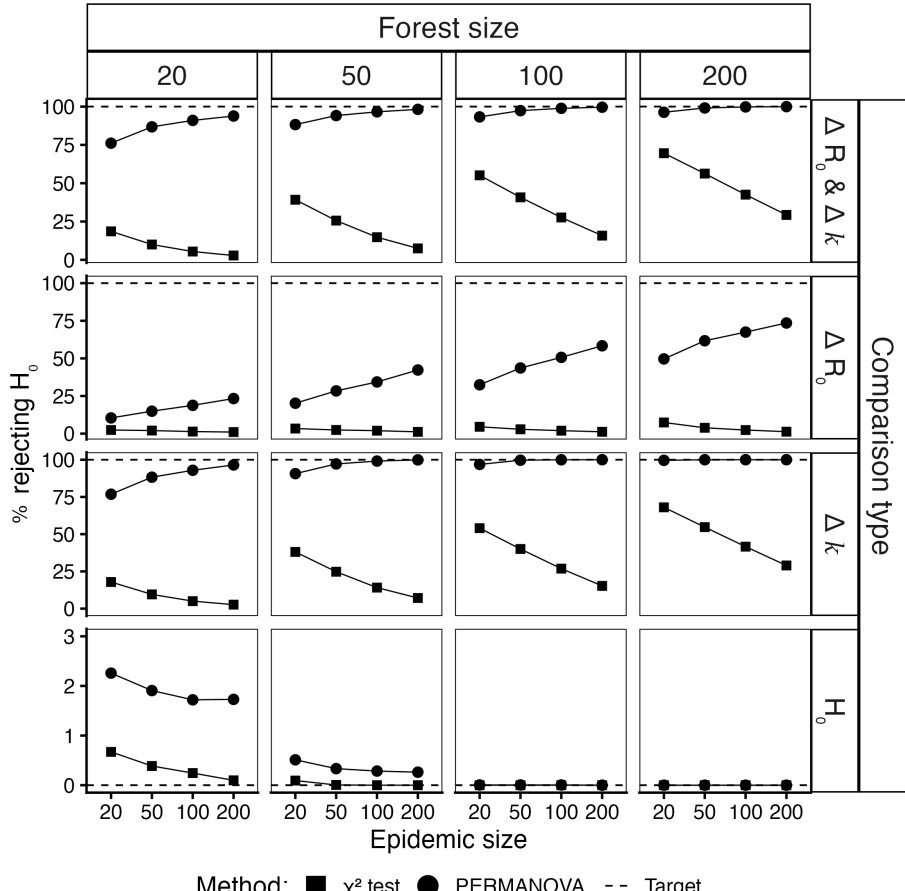

**Fig 2. Performance of $\chi^2$ and PERMANOVA for comparing epidemic forests.** This figure summarises simulation results for the $\chi^2$ test (square points) and PERMANOVA (circular points) across varied parameter conditions. The y-axis shows the percentage of tests rejecting the null hypothesis ($H_0$) of no difference between forests. The x-axis displays the epidemic sizes. Column panels refer to the forest size (*i.e.,* the number of trees in each forest). Row panels refer to the type of differences between the two forests (with $H_0$ for no differences).

establish PERMANOVA as the superior method for comparing epidemic forests when sufficient samples are available ($m \geq 100$), providing excellent sensitivity and specificity regardless of epidemic size.

## Discussion

We evaluated two statistical approaches, the chi-square ($\chi^2$) test and PERMANOVA, for distinguishing between collections of transmission trees (*i.e.,* epidemic forests) originating from different generative processes, defined by the mean and dispersion of their offspring distribution (*i.e.,* the distribution of secondary cases generated by each infected individual). The $\chi^2$ test tests for differences in the frequency of infector-infectee pairs between epidemic forests, treating each pair as isolated edges without considering their relative position within each tree. In contrast, PERMANOVA leverages customised tree-based distance metrics to quantify meaningful epidemiological differences between tree topologies, which may better signal distinct pathogen transmission dynamics.

Our simulations showed that PERMANOVA consistently outperformed the $\chi^2$ test in distinguishing epidemic forests generated under different offspring distributions. It achieved near-perfect sensitivity when forests differed in their dispersion parameter across all epidemic sizes (20–200 cases), provided forests contained at least 100 trees. However,

**Table 1. Logistic regression results for the sensitivity model (Eq 10).**

| Predictor | | Odds Ratio | *p*–value |
|---|---|---|---|
| **Intercept** | | **0.01** | **<0.001** |
| **Method** | PERMANOVA | 1.94 | <0.001 |
| **Forest size (*m*)** | 50 | 2.93 | <0.001 |
| | 100 | 6.10 | <0.001 |
| | 200 | 12.06 | <0.001 |
| **Epidemic size ($\varepsilon$)** | 50 | 0.54 | <0.001 |
| | 100 | 0.30 | <0.001 |
| | 200 | 0.15 | <0.001 |
| **Parameter difference ($\Delta$)** | | | |
| | $R_0$ | 1.70 | <0.001 |
| | $k$ | | |
| | (0, 1] vs. (0, 1] | 33.04 | <0.001 |
| | (0, 1] vs. Poisson | 36.74 | <0.001 |
| | $R_0$: $k$ | | |
| | (0, 1] vs. (0, 1] | 0.62 | <0.001 |
| | (0, 1] vs. Poisson | 0.61 | <0.001 |
| **PERMANOVA: $\varepsilon$** | | | |
| | 50 | 4.19 | <0.001 |
| | 100 | 11.85 | <0.001 |
| | 200 | 33.94 | <0.001 |
| **PERMANOVA: $\Delta$** | | | |
| | $R_0$ | 3.01 | <0.001 |
| | $k$ | | |
| | (0, 1] vs. (0, 1] | 7.77 | <0.001 |
| | (0, 1] vs. Poisson | 51.38 | <0.001 |
| | $R_0$: $k$ | | |
| | (0, 1] vs. (0, 1] | 0.21 | <0.001 |
| | (0, 1] vs. Poisson | 0.15 | <0.001 |

*Note:* Pseudo $R^2 = 0.58$ [25]. All p-values are $< 0.001$ due to the large simulation sample size ($n = 5,040,000$). ':' denotes the interaction term. The reference categories are: $\chi^2$ test for method, 20 for $\varepsilon$ and $m$, 0 for $\Delta k$.

its performance declined when forests differed solely in their mean reproduction number, especially for forests with high overdispersion (common $k < 0.5$) (supplementary S4 Fig). In such settings, the substantial stochastic variability in individual transmission masked differences in mean transmissibility. Although the $\chi^2$ test also demonstrated excellent specificity, its sensitivity was consistently lower across all scenarios and declined further as epidemic size increased. Larger epidemics produced higher forest entropy, which indicates greater variation in who infected whom across trees (supplementary S5 Fig, S1 Table). Increased entropy yielded sparse contingency tables with many low expected counts and growing degrees of freedom, which reduced the statistical power of the $\chi^2$ test (see Methods, Eq 2). In contrast, PERMANOVA became more sensitive as epidemics grew, given that additional transmission events reduced the variance in within-group distances, increasing the F-statistic (see Methods, Eq 9).

Computationally, both methods scale with epidemic size, although PERMANOVA incurs greater computational expense (see supplementary file S1 File, S2 Table). Parallelisation and constrained permutation (for PERMANOVA [26]) or replicates used in the Monte Carlo test (for $\chi^2$ test [27]) make both methods applicable to most contexts. When comparing

two forests, each with 100 trees and 100 vertices, the $\chi^2$ test takes 0.5 seconds, while PERMANOVA takes an average of 5 seconds (supplementary S2 Table). To facilitate accessibility of these methods, we have developed *mixtree* [28], a free, open-source R package available on CRAN [27]. *mixtree* implements both the $\chi^2$ test and PERMANOVA methods described in this study.

The proposed framework addresses several needs for outbreak reconstruction. First, it provides a formal approach for assessing MCMC convergence in tree space by comparing epidemic forests sampled from independent MCMC chains, which should be statistically indistinguishable when converged. This method complements existing diagnostics that focus on scalar parameter chains, which do not fully capture the complex tree structures that form the primary output of Bayesian inference models. Second, it enables rigorous comparison between competing models with different assumptions about transmission dynamics, facilitating evidence-based model selection. Third, it can detect whether incorporating additional data sources (*e.g.,* contact tracing [29]) into reconstruction efforts significantly alters the resulting transmission trees, helping researchers evaluate the value of supplementary data. However, it cannot independently determine which reconstruction is more accurate without additional validation measures.

Our study focused on comparing two forests of equal size for computational feasibility. However, both methods can compare any number of forests of varying sizes sharing the same set of vertices, as implemented in our *mixtree* package. Nonetheless, the two methods do not share identical limitations. PERMANOVA assumes full graph connectivity [30], so it cannot accommodate multiple introductions that result in disconnected trees. In contrast, the $\chi^2$ test can handle transmission trees that have multiple introductions by assigning them specific identifiers, e.g., 'Introduction A', 'Introduction B' etc. In the presence of unobserved cases, the $\chi^2$ test cannot distinguish between direct and unobserved intermediate transmissions. Importantly, PERMANOVA could be extended by modelling epidemiological, spatial or genetic distances as edge weights. For example, these weights could represent the number of infection generations between pairs of cases, thus accounting for unobserved cases. The standard graph distance used here could be replaced with a more complex metric that incorporates additional edge characteristics (*i.e.,* weights) such as the number of generations between observed cases, or the time difference between their symptoms [31] or infection dates. While our simulation framework assessed method performance when the forest's generative process differed only in its offspring distribution, other epidemic features also shape tree topology. Future work should evaluate performance under alternative assumptions about epidemic dynamics such as group transmission patterns [8], the effects of saturation [32], vaccination or new variants of concern [31], which would require developing additional distance metrics for PERMANOVA to capture such features. Our simulation framework focused on epidemics of 20–200 cases, reflecting the typical range for computational outbreak reconstruction, and our results show that PERMANOVA performs well once forests comprise 100 or more trees, corresponding to the typical effective sample size from Bayesian reconstruction models [12].

While alternative methods for comparing graph collections exist, they typically rely on abstract graph kernels not directly interpretable in our epidemiological context [33]. In contrast, our method employs a distance metric that is epidemiologically meaningful, as it corresponds to the number of generations of infection separating each pair of cases. Future work could also examine how the multivariable analysis capability of PERMANOVA may be used to quantify the relative contributions of the inference method, data type, and prior assumptions on the observed topological differences between epidemic forests. In addition to the application to epidemic reconstruction that we have considered here, this work addresses a more general methodological gap across disciplines where relational structures are represented as graphs [34–39]. In practice, diverse data sources, modelling assumptions, and analytical methods typically produce not single solutions but ensembles of plausible alternatives, *i.e.,* collections of graphs. Bayesian approaches excel at generating these collections through MCMC sampling but lack formal statistical tools for comparing the resulting posterior samples. One example of other such application area is phylogenetic tree reconstruction [39], where researchers encounter similar challenges that can lead to conflicting evolutionary hypotheses or taxonomic classifications. In information and network science, different network representations may likewise suggest distinctive social patterns or information flow dynamics.

In conclusion, our framework enables the comparison of collections of transmission trees, a special class of graph, by distinguishing meaningful structural variations from sampling and model uncertainty. We have demonstrated its utility to epidemic reconstruction, but this approach likely extends to other fields relying on graph-based representations. We encourage researchers to adapt and validate this framework to address domain-specific challenges in their respective fields, potentially developing additional metrics that capture the unique characteristics of their data structures.

## Methods

We introduce a framework for comparing collections of transmission trees, termed *epidemic forests*. We present two approaches: the first based on a $\chi^2$ test [23] on transmission pair frequencies, and the second using PERMANOVA, a method originally developed for ecological community analysis [24], on transmission tree distances. Both methods are described below and illustrated in Fig 1. We use a simulation to compare the respective performances of the two approaches.

### Epidemic forests

Transmission trees represent the spread of a disease amongst infected individuals as directed acyclic graphs (DAGs) [1]. A transmission tree $T = (V, E)$ consists of a set $V = \{v_1, v_2, \ldots, v_n\}$ containing $n$ vertices (each representing an infected individual) and a set $E = \{e_2, e_3, \ldots, e_n\}$ of $n-1$ directed edges. Each edge represents an infector-infectee pair, denoted as $e_j = (v_i, v_j)$, with $v_i, v_j \in V$ and $v_i \neq v_j$. This directed edge connects an infector $v_i$ to its infectee $v_j$, formally encoding the 'who infected whom' relationship. All vertices have an in-degree of 1, except the root, which represents the index case and has an in-degree of 0. In the absence of data to define meaningful edge weights, we assume all edges have a weight of 1.

We define an *epidemic forest* as a collection of transmission trees, each with the exact same set of vertices, but possibly different sets of edges. We consider two epidemic forests $\mathcal{F}_A = (T_1^A, \ldots, T_{m_A}^A)$ and $\mathcal{F}_B = (T_1^B, \ldots, T_{m_B}^B)$, where the $k^{\text{th}}$ tree in $\mathcal{F}_A$ is defined as $T_k^A = (V, E_k^A)$. For simplicity, we assume that the two epidemic forests have the same size ($m_A = m_B = m$), but the approaches described below can readily accommodate ($m_A \neq m_B$). In practice, an epidemic forest may be obtained by sampling from a posterior distribution via Bayesian inference (*e.g.*, MCMC) or from a stochastic transmission model [12,40].

### $\chi^2$ test

The $\chi^2$ test compares the absolute frequencies of infector-infectee pairs (*i.e.*, edges) between two epidemic forests $\mathcal{F}_A$ and $\mathcal{F}_B$. For each of the possible infector-infectee pair, we count their occurrences across all trees in a forest $\mathcal{F}_X$ as:

$$c_{ij}^{\mathcal{F}_X} = \sum_{l=1}^{m} \mathbb{1}_{((v_i, v_j) \in E_l^X)}$$

(1)

where $\mathbb{1}$ is the indicator function (yielding 1 if the pair appears in tree $T_l^X$, 0 otherwise).

The $\chi^2$ statistic for comparing forests $\mathcal{F}_A$ and $\mathcal{F}_B$ is:

$$\chi^2 = \sum_{(i,j) \in \mathcal{P}} \frac{(c_{ij}^{\mathcal{F}_A} - c_{ij}^{\mathcal{F}_B})^2}{c_{ij}^{\mathcal{F}_A} + c_{ij}^{\mathcal{F}_B}}$$

(2)

where $\mathcal{P} = \{(i, j) \mid i \neq j, c_{ij}^{\mathcal{F}_A} + c_{ij}^{\mathcal{F}_B} > 0\}$ includes only infector-infectee pairs observed in at least one forest. Under the null hypothesis that both forests stem from the same underlying frequency distribution of infector-infectee pairs, $\chi^2$ follows a chi-square distribution with $|\mathcal{P}| - 1$ degrees of freedom, where $|\mathcal{P}|$ denotes the number of unique infector-infectee pairs

observed. To accommodate small counts, the non-parametric Monte Carlo version of the chi-square test (999 replicates) was then used [23,41]. This formulation assumes equal forest sizes ($m_A = m_B = m$). Under the null hypothesis that both forests are sampled from the same distribution of infector-infectee pairs, the expected count for pair $(i,j)$ in forest $\mathcal{F}_A$ is $E_{ij}^{\mathcal{F}_A} = \frac{c_{ij}^{\mathcal{F}_A} + c_{ij}^{\mathcal{F}_B}}{2}$, and similarly for $\mathcal{F}_B$. Substituting these expected values into the classical chi-squared formula $\frac{(O-E)^2}{E}$ and simplifying yields Equation 2. When forest sizes differ ($m_A \neq m_B$), the standard chi-squared formulation applies, with expected counts proportional to forest size. This generalisation is implemented in the `mixtree` package.

## PERMANOVA

PERMANOVA is a generic approach used to test group differences using pairwise distances between all observations of a sample and makes no model assumptions [24]. Here, we apply it to test whether distances between transmission trees differ when the trees belong to the same epidemic forest versus different forests.

**Distance between two transmission trees.** The field of phylogenetics offers a range of established methods for comparing tree structures, providing several distance metrics for quantifying topological differences between pairs of phylogenies [18,42–46]. These methods typically follow a two-step process: (i) convert trees into vectors of pairwise distances between all sampled taxa and (ii) compute Euclidean distances between these vectors.

A commonly used metric for the first step is the *patristic* distance [44], defined as the sum of branch lengths on the path separating two taxa, reflecting the evolutionary distance between them. Adapting this concept to transmission trees, we define the graph distance between cases (*i.e.*, vertices) $v_i$ and $v_j$ as the sum of edge weights along their connecting path on the undirected graph. Since all edges here have a weight of 1, this distance directly corresponds to the number of transmission events between cases, carrying clear epidemiological meaning. An illustration of graph distances in a transmission tree is available in the supplementary material (S1 Fig).

We denote $\pi(.)$ the function mapping a transmission tree $T$ of size $n$ into a vector of $\frac{n(n-1)}{2}$ graph distances:

$$\mathbf{d}_T = \pi(T) \tag{3}$$

where $\mathbf{d}_T \in \mathbb{R}_+^{n(n-1)/2}$.

The dissimilarity between two trees $T_k$ and $T_l$ is then quantified by the Euclidean distance between the respective vectors of graph distances, calculated as the norm:

$$D(T_k, T_l) = \|\mathbf{d}_{T_k} - \mathbf{d}_{T_l}\| \tag{4}$$

This distance captures topological differences by evaluating how the relative positions of vertices, encoded as graph distances, diverge between the two trees. If $T_k$ and $T_l$ have identical edge sets, their graph distance matrices are equal, yielding $D(T_k, T_l) = 0$; otherwise, discrepancies in path lengths increase the distance.

**Outline of the method.** Given two epidemic forests, $\mathcal{F}_A$ and $\mathcal{F}_B$, each containing $m$ transmission trees, we apply PERMANOVA to test whether tree topologies differ significantly between forests. Broadly, the method partitions pairwise distances between all trees into within-group ($\text{SS}_W$) and between-group ($\text{SS}_B$) components [24], based on pre-defined groups (here, the two forests). Statistical significance is assessed through permutation testing, where forest identifiers (e.g., '$\mathcal{F}_A$', '$\mathcal{F}_B$') are randomly reassigned multiple times.

We define the combined epidemic forest as $\mathcal{F}_{A \cup B} = \mathcal{F}_A \bigcup \mathcal{F}_B$, containing all trees from $\mathcal{F}_A$ and $\mathcal{F}_B$. The total sum of squares, $\text{SS}_T$, representing the overall variance across all trees in $\mathcal{F}_{A \cup B}$, is:

$$\text{SS}_T = \frac{1}{2m} \sum_{k=1}^{2m} \sum_{l=1}^{2m} D(T_k^{A \cup B}, T_l^{A \cup B})^2 \tag{5}$$

The double summation computes squared pairwise distances amongst the $2m$ trees in $\mathcal{F}_{\mathcal{A}\bigcup\mathcal{B}}$, which decomposes to:

$$SS_T = \frac{1}{2m}\left(\sum_{k=1}^{m}\sum_{l=1}^{m}D(T_k^A, T_l^A)^2 + \sum_{k=1}^{m}\sum_{l=1}^{m}D(T_k^B, T_l^B)^2 + 2\sum_{k=1}^{m}\sum_{l=1}^{m}D(T_k^A, T_l^B)^2\right)$$

(6)

The within-group sum of squares $SS_W$ measures the variance within forests:

$$SS_W = \frac{1}{m}\left(\sum_{k=1}^{m}\sum_{l=1}^{m}D(T_k^A, T_l^A)^2 + \sum_{k=1}^{m}\sum_{l=1}^{m}D(T_k^B, T_l^B)^2\right)$$

(7)

where each term sums the squared distances among all pairs within each forest, normalised by $m$. The between-group sum of squares ($SS_B$), capturing variability between the forests, is:

$$SS_B = SS_T - SS_W$$

(8)

The PERMANOVA test statistic [24] is:

$$F = \frac{SS_B}{SS_W/(2m-2)}$$

(9)

The reference distribution of $F$ under the null hypothesis of no differences between groups is generated by a Monte Carlo procedure where forests' identifiers are permuted a large number of times (*i.e.,* 999 by default). *p*-values are calculated as the proportion of permuted $F$-values exceeding the observed $F$ [24].

**Simulation study**

We conducted a simulation study to evaluate the performance of the $\chi^2$ test and PERMANOVA in distinguishing between simulated epidemic forests drawn from distinct generative processes corresponding to different epidemic dynamics. The simulation framework is illustrated in Fig 3.

1. We simulate a *reference* transmission tree $\mathcal{T}$ with $\varepsilon$ infections from offspring distribution NegBin($R_0$, $k$). This process is repeated 100 times to account for the stochasticity of epidemic dynamics.

2. We generate reconstructed forests $\mathcal{F}_A$ and $\mathcal{F}_B$, each containing $m$ trees, by re-assigning infector-infectee relationships from NegBin($R_{0,A}'$, $k_A'$) and NegBin($R_{0,B}'$, $k_B'$), conditional on $\mathcal{T}$'s dates of infection and case identifiers. In this example, $\mathcal{F}_A$ = NegBin($R_0 = 2$, $k = 0.1$) and $\mathcal{F}_B$ = NegBin($R_0 = 2$, $k = 10$).

3. The $\chi^2$ test and PERMANOVA are applied to test whether the two epidemic forests stem from the same generative process.

   **Simulating epidemic forests.**  We generated epidemic forests through a three-stage process to systematically evaluate forest comparison methods across diverse transmission scenarios.
   First, we defined the parameter space for the simulations:

- **Epidemic size**: $\varepsilon \in \{20, 50, 100, 200\}$. The number of infected individuals, corresponding to the number of vertices in the tree.

- **Basic reproduction number**: $R_0 \in \{1.5, 2, 3\}$. The mean number of secondary infections per case in a fully susceptible population, corresponding to the mean of the negative binomial offspring distribution.

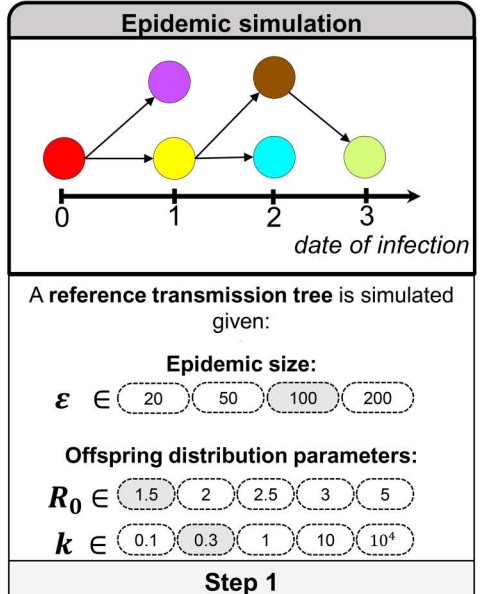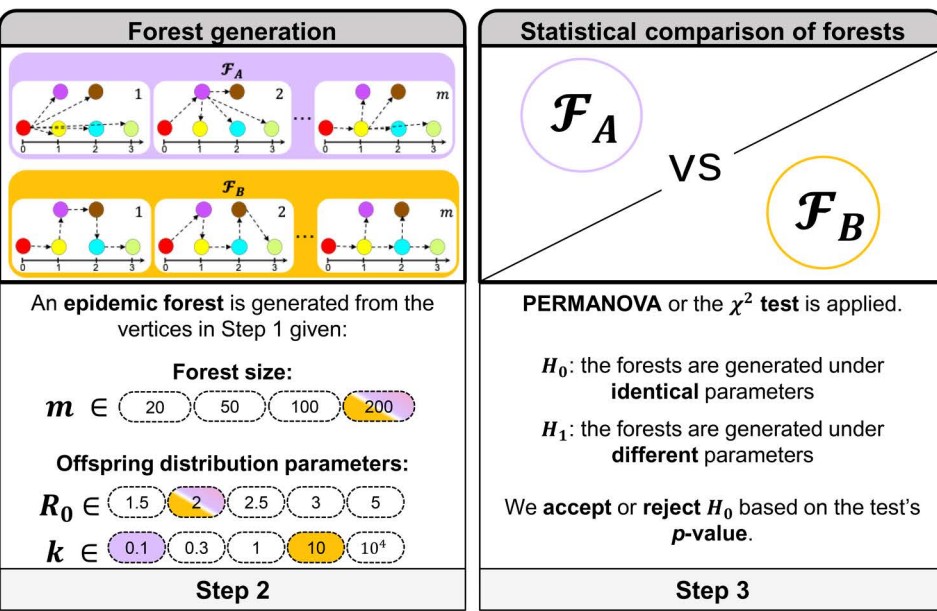

**Fig 3. Simulation framework for assessing the performance of the $\chi^2$ test and PERMANOVA.** Diagram illustrating the simulation study to assess the respective performances of $\chi^2$ test and PERMANOVA for detecting differences between pairs of epidemic forests.

- **Dispersion parameter**: $k \in \{0.1, 0.3, 0.5, 1, \infty\}$. Controls heterogeneity in individual transmission, corresponding to the dispersion of the negative binomial offspring distribution. Lower values indicate greater overdispersion; as $k \to \infty$, the distribution converges to Poisson.

For each epidemic sizes $\varepsilon$, we defined offspring distributions NegBin($R_0$, $k$) using all pairwise combinations of basic reproduction number $R_0$ and dispersion parameter $k$.

Second, we generated *reference* transmission trees. For each parameter combination ($\varepsilon, R_0, k$) we simulated a reference transmission tree $\mathcal{T}^\varepsilon_{(R_0,k)}$ using a stochastic branching process. Secondary infections per case were drawn from NegBin($R_0$, $k$), and generation times followed a gamma distribution with a mean of 12 days and standard deviation of 6 days (see supplementary S1 File). We generated 100 replicate trees per parameter set to account for stochasticity. Simulations were initialised with 10,000 susceptible individuals, ran for a maximum of 365 days, and terminated upon reaching exactly $\varepsilon$ infections, thereby excluding saturation effects. Within each reference tree $\mathcal{T}^\varepsilon_{(R_0,k)}$, infected individuals were assigned identifiers $v \in \{1, \ldots, \varepsilon\}$, ordered by their dates of infection $t_v$.

Third, we constructed epidemic forests by reassigning cases' ancestries. For each reference tree $\mathcal{T}^\varepsilon_{(R_0,k)}$, we generated forests $\mathcal{F}_{\mathcal{T}^\varepsilon_{(R_0,k)}}(R'_0, k')$ by conditioning on the observed infection set $\mathcal{I}_{\mathcal{T}^\varepsilon_{(R_0,k)}} = \{(v, t_v)\}^\varepsilon_{v=1}$ while resampling ancestries from NegBin($R'_0, k'$) (see supplementary S1 File). Each forest comprised $m = 200$ trees. This procedure yielded 15 distinct forests per reference tree (one for each offspring distribution pair ($R'_0, k'$)), including one forest matched the reference tree's generative process, where ($R'_0, k'$) = ($R_0, k$).

This procedure generated a total of 6,000 reference trees ($|\varepsilon| \times |R_0| \times |k| \times$ replicates $= 4 \times 3 \times 5 \times 100$), each generating 15 distinct forests ($|R'_0| \times |k'|$), yielding 120 pairwise forest comparisons per reference tree ($\binom{15}{2} + 15$), resulting in a total of 720,000 forest comparisons.

**Assessing statistical performance.** For each of the 720,000 forest comparisons ($\mathcal{F}_A$ vs. $\mathcal{F}_B$), we performed the $\chi^2$ test and PERMANOVA under 4 forest sizes $m \in 20, 50, 100, 200$, where $m$ denotes the number of trees sampled from each forest. This resulted in a total of 5,760,000 tests performed. For each parameter combination, we measured:

- **Sensitivity**: The proportion of tests that correctly rejected the null hypothesis ($H_0$) when comparing forests generated with different offspring distributions, *i.e.*, $\mathcal{F}_{\mathcal{T}}(R_0, k)$ vs. $\mathcal{F}_{\mathcal{T}}(R'_0, k')$ where $(R_0, k) \neq (R'_0, k')$.

- **Specificity**: The proportion of tests that correctly accepted $H_0$ when comparing forests generated with identical offspring distributions, *i.e.*, $(R_0, k) = (R'_0, k')$.

To quantify the factors influencing test sensitivity, we fit a logistic regression model to all comparisons where forests were generated under different parameter settings ($H_1$; $n = 5{,}040{,}000$). The binary outcome was whether the test correctly rejected the null hypothesis ($H_0$). We compared four nested models using the Akaike Information Criterion and selected the model with the lowest value. The final model included main effects for statistical method (PERMANOVA or $\chi^2$), forest size ($m$), epidemic size ($\varepsilon$), and parameter differences between forests ($\Delta R_0$ and $\Delta k$). It also included all two-way and three-way interaction terms involving the method:

$$\begin{aligned}
\mathrm{logit}(P(\text{reject } H_0)) = {} & \beta_0 + \beta_{\mathrm{method}} + \beta_m + \beta_\varepsilon + \beta_{\Delta R_0} + \beta_{\Delta k} \\
& + \beta_{\mathrm{method}:\varepsilon} + \beta_{\mathrm{method}:\Delta R_0} + \beta_{\mathrm{method}:\Delta k} \\
& + \beta_{\Delta R_0:\Delta k} + \beta_{\mathrm{method}:\Delta R_0:\Delta k} + e
\end{aligned}$$

(10)

Where ':' represents the interaction term and $e$ is the normally distributed residuals. Results are reported as odds ratios using the $\chi^2$ test, the smallest forest size ($m = 20$), the smallest epidemic size ($\varepsilon = 20$), and no difference in dispersion ($\Delta k = 0$) as reference categories. The model achieved a pseudo $R^2$ of 0.58 [25].

Both methods achieved near-perfect specificity (>97%) across all conditions, precluding regression analysis.

## Supporting information

**S1 File. Description of the methods for generating epidemic** forests.
(PDF)

**S1 Fig. Calculation of graph distances in a transmission tree.** We define the graph distance between two vertices as the number of edges on the path that connects them. Epidemiologically, this corresponds to the number of transmission events that separate these two cases. A. A transmission tree with 5 cases. The coloured dashed lines show the unique paths connecting case 5 to all other cases. B. The matrix representation of graph distances between all pairs of cases. The coloured numbers correspond to the number of transmission events between case 5 and all other cases, matching the coloured paths shown in A. For illustration, we only represent the lower triangle but the matrix is symmetric.
(TIF)

**S2 Fig. Performance of $\chi^2$ test and PERMANOVA for distinguishing epidemic forests.** Median p-values and inter-quartile ranges for the $\chi^2$ test (squares) and PERMANOVA (circles) across epidemic sizes (x-axis), forest sizes (columns), and parameter conditions (rows). Grey shading indicates desired p-value ranges: below $\alpha = 0.05$ when forests differ in at least one parameter (rows 1–3, reject $H_0$) and above $\alpha = 0.05$ when forests share identical parameters (row 4, accept $H_0$).
(EPS)

**S3 Fig. Performance of $\chi^2$ and PERMANOVA in distinguishing epidemic forests.** Each panel shows the proportion of tests rejecting the null hypothesis ($p < 0.05$) when comparing epidemic forest $\mathcal{F}_{\mathcal{A}}$ and $\mathcal{F}_{\mathcal{B}}$. The upper triangle shows the $\chi^2$ test results; the lower triangle shows PERMANOVA results. Outer columns refer to epidemic size ($\varepsilon$), common to both forest. Forests can differ in their offspring distribution parameter: $R_0$ (inner columns) and $k$ (x and y axes; x-axis labels omitted for clarity, values identical to the y-axis). Red diagonal lines indicate comparisons where both forests share identical parameters ($H_0$ true; low rejection rates indicate good specificity). The other cells compare forests with different

parameters (high rejection rates indicate good sensitivity). Both methods maintain excellent specificity (diagonal), but PERMANOVA demonstrates superior sensitivity.
(EPS)

**S4 Fig. Sensitivity of PERMANOVA when comparing epidemic forests that differ only in $R_0$.**Sensitivity (y-axis) is the proportion of tests correctly rejecting the null hypothesis when comparing forests of 200 trees generated with different $R_0$ ($\Delta R_0$, colour) but identical $k$ (x-axis). Columns correspond to epidemic size.
(EPS)

**S5 Fig. Variation in infector-infectee relationships across epidemic forests** Histogram of mean scaled entropy across epidemic forests ($m = 200$ trees per forest), stratified by simulation parameters.For each infectee $j$, the scaled entropy $H_j$ (x axis) quantifies variation in their assigned infector across all trees in a forest, computed using the normalised Shannon entropy formula [49]: $H_j = \frac{-\sum_{i=1}^{K_j} p_{ij} \log(p_{ij})}{\log(K_j)}$, where $p_{ij}$ is the proportion of trees in which individual $i$ infects $j$, and $K_j$ is the number of distinct infectors of $j$ observed across the forest. Values range from 0 (identical infector in all trees) to 1 (all possible infectors equally frequent). The mean scaled entropy ($\bar{H}$) is obtained by averaging $H_j$ over all cases. Columns refer to epidemic sizes ($\varepsilon$), rows refer to the mean reproduction number $R_0$ and dispersion parameter $k$ of the negative binomial offspring distribution. Average entropy for our simulations is 77% and increases with epidemic size, due to greater variation in infector assignment (S1 Table).
(EPS)

**S1 Table. Linear regression results for the mean scaled entropy model.**The mean scaled entropy $\bar{H}$ for each forest was modelled as a linear function of epidemic size, reproduction number, and dispersion. $\bar{H}^* = \beta_0 + \beta_\varepsilon + \beta_{R_0} + \beta_k + \epsilon$ (11) where $\beta_0$ is the intercept, each $\beta$ term represents the categorical effect of epidemic size ($\varepsilon$), reproduction number ($R_0$), and dispersion parameter ($k$) respectively, and $\epsilon$ is the residual error. The model explained 68.2% of the variance ($R^2 = 0.682$) in mean scaled entropy across 90,000 simulated forests, with coefficient estimates shown in supplementary S1 Table.
(TEX)

**S6 Fig. ROC curves for the $\chi^2$ test and PERMANOVA** Each panel shows the Receiver Operating Characteristic (ROC) curves plotting true positive rate (sensitivity) against false positive rate (1-specificity) for the $\chi^2$ test (blue) and PERMANOVA (orange) across all simulations for all possible significance thresholds ($0 \leq \alpha \leq 1$).Panels are arranged by epidemic size (columns: 20–200 cases) and forest size (rows: 20–200 trees), x-axis tick labels are omitted for clarity, as both axes share the same scale.
(EPS)

**S7 Fig. Area under curve (AUC) for $\chi^2$ test and PERMANOVA** This figure shows the AUC derived from the Receiver Operating Characteristic (ROC) curves (supplementary S6 Fig) of the two tests evaluated in our simulations.The y-axis displays the AUC value, with higher values corresponding to better performances. An AUC of 1 corresponds to a test with perfect sensitivity and specificity. The x-axis displays the epidemic size, i.e., the number of cases in the simulated epidemics, while panels refer to the forest size, i.e., the number of trees in each forest.
(EPS)

**S2 Table. Benchmark results of execution times in seconds for the $\chi^2$ test and PERMANOVA, comparing two epidemic forests with 100 trees and 100 vertices each.**Both tests used 999 permutations (PERMANOVA) / Monte Carlo replicates ($\chi^2$ test) without parallelisation and were replicated 100 times per method. For the $\chi^2$ test, we compute the frequency of each infector-infectee pair across all trees between forests. In the worst-case scenario, where every possible infector-infectee pair (*i.e., $n(n-1)$* pairs) appears at least once in either forest, the computational time for the $\chi^2$

test increases with the number of trees in each forest ($m$) and the square of the number of cases ($n^2$), since it considers all infector-infectee pairs for every tree (See Methods, Fig 1). Therefore the overall computational time will increase as a function of $mn^2$. On the other hand, PERMANOVA involves a two-step process. First, it computes pairwise distances between all vertices within each tree (supplementary S1 Fig), which scales as a function of $n^2$ for a given tree. Second, it calculates pairwise distances between all trees (Fig 1), which scales as a function of $m^2$. Therefore the overall computational time will increase as a function of $m^2 n^2$.
(TEX)

## Author contributions

**Conceptualization:** Cyril Geismar, Anne Cori, Thibaut Jombart.

**Data curation:** Cyril Geismar.

**Formal analysis:** Cyril Geismar.

**Funding acquisition:** Peter J. White, Anne Cori, Thibaut Jombart.

**Methodology:** Cyril Geismar, Anne Cori, Thibaut Jombart.

**Project administration:** Cyril Geismar.

**Resources:** Peter J. White.

**Software:** Cyril Geismar.

**Supervision:** Anne Cori, Thibaut Jombart.

**Validation:** Cyril Geismar, Anne Cori.

**Visualization:** Cyril Geismar.

**Writing – original draft:** Cyril Geismar.

**Writing – review & editing:** Cyril Geismar, Anne Cori, Thibaut Jombart.

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
