## [Decision Letter · Decision Letter 0]

22 Jun 2025

PCOMPBIOL-D-25-00809

A statistical framework for comparing epidemic forests

PLOS Computational Biology

Dear Dr. Geismar,

Thank you for submitting your manuscript to PLOS Computational Biology. After careful consideration, we feel that it has merit but does not fully meet PLOS Computational Biology's publication criteria as it currently stands. Therefore, we invite you to submit a revised version of the manuscript that addresses the points raised during the review process.

Please submit your revised manuscript within 60 days Aug 22 2025 11:59PM. If you will need more time than this to complete your revisions, please reply to this message or contact the journal office at ploscompbiol@plos.org. Please include the following items when submitting your revised manuscript:

We look forward to receiving your revised manuscript.

Kind regards,

Sasikiran Kandula

Academic Editor

PLOS Computational Biology

Jennifer Flegg

Section Editor

PLOS Computational Biology

**Additional Editor Comments:**

We received detailed feedback from three reviewers. The consensus recommendation is a ‘major revision’ which I agree with.

Broadly, Reviewer 1 has asked for an important clarification question and strongly suggests additional simulation exercises. Reviewer 2 would like elaboration and discussion of some of the theoretical grounding and assumptions. Reviewer 3 (co-reviewed) doubts that some of the statements made by the authors are sufficiently supported by evidence presented and recommends specific tests.

I think these are substantive concerns and request the authors to consider expanding the manuscript in light of these recommendations.

**Journal Requirements:**

At this stage, the following Authors/Authors require contributions: Cyril Geismar, Anne Cori, and Thibaut Jombart. Please ensure that the full contributions of each author are acknowledged in the "Add/Edit/Remove Authors" section of our submission form.

4) We notice that your supplementary Figures, and Tables are included in the manuscript file. Please remove them and upload them with the file type 'Supporting Information'. Please ensure that each Supporting Information file has a legend listed in the manuscript after the references list.

Potential Copyright Issues:

i) Figure 4. Please confirm whether you drew the images / clip-art within the figure panels by hand. If you did not draw the images, please provide (a) a link to the source of the images or icons and their license / terms of use; or (b) written permission from the copyright holder to publish the images or icons under our CC BY 4.0 license. Alternatively, you may replace the images with open source alternatives. See these open source resources you may use to replace images / clip-art:

**Reviewers' comments:**

Reviewer's Responses to Questions

**Comments to the Authors:**

Reviewer #1: The paper proposes statistical methods for the comparison of large groups of epidemic transmission trees, using either a chi-squared test or a non-parametric version of a multivariate analysis of variance test (PERMANOVA). This is a useful contribution to the field and definitely has a potential home in the literature. The paper is concise and clear. I have a major concern about the approach which needs addressing, and I also think that further simulations are strongly warranted.

What really concerns me is whether the difference in degree distribution is what drives the dissimilarity of the forests at all. The vertices here are labelled, and both methods use the labels. The chi-squared version has the usual contingency table, and PERMANOVA uses an ordered vector of distances. Each node in one set of forests must have a twin in the other. How are these twins assigned? It seems to me that the answer is “arbitrarily”. Unless this is poorly explained, each comparison is made between forests all of which have the same underlying true tree (p14). If this is not true and members of the same forest can have a different true tree, then my concern goes away. But if not, if nodes i and j in one tree are a transmission pair in the tree underlying forest A, they may be distantly related in the one generating forest B. The vector or matrix is going to be very different even if the underlying mathematical model is the same. In this case, does this test anything other than that if true trees are different? If not, why change the generating model at all?

That aside, the other problem I have with this presentation is that the simulation exercise is much too rudimentary and only scratches the surface of the possible applications. It proposes many of those applications without exploring them, and since this is not a long piece of work in its current iteration, I think there is considerable scope for that exploration.

For example:

• The issue of unsampled cases is raised briefly on page 9 but not explored. Could you see how well this works to identify if two sets of trees come from epidemics of different sizes but with the same number of observed nodes?

• You stop the simulation at 100 days (p20). Another use case would be to see if this can discriminate between finished and ongoing epidemics with the same number of observed nodes.

• The number of different generating models you could compare is effectively infinite, but using one comparison only does feel inadequate.

Minor points:

Please put line numbers in a revised version.

Page 2:

Paragraph 2: Another reason for uncertainty (perhaps the key one) is the lack of discriminatory power in choosing between possible transmission pairs (the subject of the Campbell et paper in the genomics realm).

Paragraph 3: “…determining whether differences between them represent meaningful variations in transmission dynamics or are due to sampling and model uncertainty is challenging.” Do you mean MCMC sampling or epidemic sampling? If the latter, I do not think this paper shows how its method helps.

Paragraph 4: MCMC diagnostics are not used to compare competing approaches; if the diagnostics suggest a problem, the analysis has failed before you get to the comparison. It’s not clear what “model reliability” means here but the same comment could also apply.

Page 3:

Paragraph 1: I do not see how consensus trees are “abstract representations” any more than actual transmission trees are, and if your consensus tree has loops, it is clearly produced using a bad algorithm. The maximum parent credibility and Edmonds trees are used in the papers that are cited here!

Paragraph 3: Why is it important that PERMANOVA is multivariate when the use case here is univariate?

Page 5:

Paragraph 1: 200 cases is quite small given how fast the method seems to work; I would like to see this extended into the thousands.

Page 9:

Paragraph 1: Could you not handle multiple introductions in PERMANOVA by very heavily weighting the distance between the root nodes of each introduction?

It is not clear how figure 1 illustrates this dedicated category.

Is using different variables as edge weights in the same analysis how you could make use of PERMANOVA being multivariate?

Paragraph 2: I found it a bit difficult to believe that this addresses a profound gap in the general graph theory literature, and a bit of searching turned up the Python GTST package for a start. There is existing work outside epidemiology. It’s also hard to see how exactly this would be of use in phylogenetic trees; there are developed methods for that and I cannot see the new insight this brings.

Page 10:

Regarding the methods generally: I see no immediate reason why the assumptions underlying these procedures are not met (e.g. PERMANOVA has no assumptions of independence, which is good because trees would strongly violate them) but I think this should be spelled out.

Page 11:

Paragraph 3: “Patristic distance” is the wrong term. It is so named because of notions of shared ancestry (and even this was perhaps eccentric - look up “patristic” in the dictionary!) and that does not really make sense here. This is just bog standard graph distance.

Paragraph 4: Typo (n-1)/2 should be n(n-1)/2

P12

Paragraph 5: Typo “numbe”

Reviewer #2: The paper addresses an interesting problem, namely inference where the parameter is a tree (graph).

The possible space of trees on N distinguishable nodes is huge and some inferential procedure based on various kinds of data puts a probability distribution (a posteriori, usually) on this space.

Various classical inferential problems then arise: what is a confidence set in this space, how to test equality of two distributions, etc... the information that can be obtained about the distribution is usually simulation based...

In the paper, the theoretical groundwork is somewhat missing... For instance, what is an epidemic forest, said to be a collection of plausible trees? Is it a confidence set? What are defining properties of this forest?

The same lack of precision affects the aim of differentiating two epidemic forests.. is this a test about the possible difference of the underlying measures on the tree space?

Already, talking about obtaining samples from an a posteriori distribution through MCMC brings the problem of dependence/approximate independence between the obtained trees...

The authors present two ad hoc methods to compare two epidemic forests. Both methods are based on reasonable functionals of the trees, but no discussion about other possibilities is presented.

A detailed critique of this spread reconstruction is the small role given in the paper to multiple introductions and missing cases, which are real problems, but avoided in this analysis of the proposed methods. To be honest, this may be a problem also of the models generating the trees that may not allow such complications...

In conclusion, the paper presents a useful ad hoc addition to methods that can be applied to inference yielding trees, but personally I would have appreciated a little bit more theoretical analysis of the underlying inferential problem...

Reviewer #3: This manuscript addresses the challenge of developing an approach to compare ensembles of transmission trees, which the authors call ‘epidemic forests’. This problem arises naturally in comparing outputs from Bayesian packages for transmission tree reconstruction, though the authors later argue that the methods could have broader application to phylogenetics or other fields. They propose two methods, one based on a chi-square test and one based on PERMANOVA, and evaluate the performance of the tests on simulated data. In particular, they constructed pairs of forests that differed by a tunable amount, and tested the sensitivity and specificity of the two methods to correctly discern the difference across a range of forest sizes and tree sizes. They conclude that the PERMANOVA approach is superior across almost all circumstances, except for quite small forests and relatively small tree sizes where the chi-square approach is better. They make these methods available in an R package.

I found the manuscript very clearly written, with concise text and clean lines of logic throughout. The problem is described and motivated well and the analysis and results are mostly very well explained. The manuscript is quite brief but conveys the concepts well and draws a fairly clear conclusion. I think these will be useful tools for the community working on MCMC reconstruction of transmission trees, and possibly for other communities as well.

However, the brevity of the manuscript also leads to shortcomings, in terms of how convincing the analyses are, and how thoroughly they are discussed. I did not feel that the broad and confident conclusions were supported by the evidence presented, and was not convinced by a key aspect of the methods. I summarize my concerns below.

1. The largest issue is that the authors draw broad conclusions about the relative performance of the two methods, but this is based on a quite limited analysis of simulated data that differ in a single dimension. The simulation leads to two ‘source’ forests G_Pois and G_NegBin, which are generated by a two-step process of simulating two original trees under very different assumptions about the offspring distribution, and then generating a forest from each of them by attempting reconstruction using a separate R package, outbreaker2. Pairs of ‘test’ forests are then drawn from these two source forests – one as a straight subsample (with replacement) from G_NegBin, and the other as a weighted mixture of draws from G_Pois and G_NegBin. The weighting parameter (called the overlap frequency) is then a measure of how similar the two test forests are, which enables the authors to assess sensitivity and specificity of their forest comparison methods. There are several issues here:

a) The mixed forest arising from this construction is a curious hybrid, comprised of a discrete mixture of two very different types of tree. While such a mixture could arise (e.g. if an MCMC chain is hopping between two distinct solutions in tree-space), it is certainly not a general model for two forests that differ from each other – especially since one of the ‘types of tree’ is drawn from the same distribution as the forest we are comparing to. To be blunt, this feels like a peculiar special case contrived to yield a clean series of comparisons, but likely not very relevant to real-world comparisons.

b) A more natural comparison would be between two forests that drift continuously away from each other in tree space (i.e. analogous to two unimodal distributions, rather than a unimodal vs a bimodal mixture distribution). This could be accomplished in the current context by constructing a series of forests where the negative binomial k takes a series of values, e.g. 0.1, 0.3, 1, 3, Poisson. Using this approach, the authors could still generate a series of comparisons that range from ‘no difference’ to ‘k=0.1 vs Poisson’, but the distributions of trees within each forest would be more coherent. I suspect this would be more relevant to most forest comparison problems that arise in research – or in any case, it is a different test to the mixture distribution in the current ms and would test the generality of the conclusions.

c) Even so, the comparisons in the study would be along a single dimension (k) for fixed values of other parameters (notably R). It would bolster the generality of the conclusions to test another dimension, either by replicating the k series for a few different R values, or by exploring an R series instead.

d) If the authors believe the current series based on the mixture distribution and overlap parameter is valid, then another route toward generalization would be to show results for that and a parameter series.

2. The current ms draws the strong conclusion that PERMANOVA outperforms the chi-square test in almost all contexts, and states some numerical thresholds for epidemic sizes and forest sizes where this statement holds true. This is valid for the results shown, but there is no attempt to test the conclusions (or thresholds) more generally, and no caveats are stated. I recognize that there are limits of space and complexity in how many results can be shown, but either the results should be tested soundly for generality or clear caveats need to be added at appropriate points throughout the ms.

3. In the current simulation scheme, differing degrees of ‘contact tracing information’ are provided to the outbreaker2 runs for the negative binomial and Poisson cases. The authors state that this was tuned empirically to achieve reconstruction accuracy from 60-80% across epidemic sizes. This initially seemed reasonable, but upon reflection I realized that this is a hidden ‘fudge factor’ that will shape the quantitative results of the method comparison, since it will shape the ‘within-forest variance’ in tree structure which has direct bearing on the PERMANOVA method. How would results differ if the authors had chosen 50% reconstruction accuracy instead? Or 90%?

4. In the expression for the chi-squared statistic (equation 2), I was not clear how the denominator was chosen. The classical chi-squared statistic has terms (O – E)^2/ E. While I can see that neither of the c_ij parameters is the ‘Expected’ value, they will have roughly equal magnitude and so the statistic as written seems roughly equivalent to adding a factor 2 to the denominator. My clever graduate student (who co-reviewed this ms with me) then came up with a derivation where the factor of 2 canceled out, which may explain the form presented in the ms, but in any case some more explanation is needed.

5. The simulation approach for the branching processes is unconventional, but should be roughly correct if S(0) is much greater than the maximum outbreak size. But the value of S(0) is not stated. (I’m not too concerned about this, since the authors’ approach does not hinge on perfect branching process simulation – it just needs a systematic way to generate different trees. But since it is all posed in branching process terms, it would be good to be accurate to avoid confusing readers.)

6. The abstract makes a bold claim that both methods have ‘near-perfect sensitivity’, but then in the Results it becomes clear that this is true only when the overlap parameter is less than 0.6. This is a very important detail – sensitivity is perfect IF the forests are sufficiently different.

7. In the discussion, the authors state that the methods can be applied more generally to any problem of comparing collections of graphs. Shouldn’t there be a restriction to acyclic graphs? There is no unique way to compute patristic distances on a cyclic graph. Or if the authors are envisioning a different approach to generating the distance matrices, then please clarify.

8. Some of the color choices in Figure 1 could be more d

**Have the authors made all data and (if applicable) computational code underlying the findings in their manuscript fully available?**

Reviewer #1: **No:** The simulated data does not seem to be available

Reviewer #2: Yes

Reviewer #3: Yes

PLOS authors have the option to publish the peer review history of their article (what does this mean?). If published, this will include your full peer review and any attached files.

Reviewer #1: No

Reviewer #2: No

Reviewer #3: **Yes:** Jamie Lloyd-Smith

**Figure resubmission:**
---

## [Decision Letter · Decision Letter 1]

22 Jan 2026

PCOMPBIOL-D-25-00809R1

A statistical framework for comparing epidemic forests

PLOS Computational Biology

Dear Dr. Geismar,

Thank you for submitting your manuscript to PLOS Computational Biology. After careful consideration, we feel that it has merit but does not fully meet PLOS Computational Biology's publication criteria as it currently stands. Therefore, we invite you to submit a revised version of the manuscript that addresses the points raised during the review process.

We look forward to receiving your revised manuscript.

Kind regards,

Jennifer A. Flegg

Section Editor

PLOS Computational Biology

Jennifer Flegg

Section Editor

PLOS Computational Biology

**Journal Requirements:**

1)  Thank you for including the following in your data availability statement: 'The resulting data is stored on a Zenodo archive: 10.5281/zenodo.17704455.'  However, the accession number provided is currently inaccessible. Please verify and provide an active link to facilitate access to the data.

**Reviewers' comments:**

Reviewer's Responses to Questions

**Comments to the Authors:**

Reviewer #1: I thank the authors for the revisions. The new version begs considerably fewer questions and answers my concerns well for the most part. One outstanding matter is that text S1 really just asserts that the offspring distribution remains the same once the parents are permuted. I would like to see this demonstrated, either in simulations or analytically.

Minor points:

L123: Type “test tests”

L176: Perhaps elaborate on “assigning them to a dedicated category in the edge list”. Do you mean you just pre-specify which introduction each case belongs to?

L260: It’s previously been claimed that it’s easy to expand to different forest sizes but then it’s just said that equal sizes are assumed here.

L299: “Forest labels” needs a bit of elaboration

The ordering of supplementary files seems off

Text S1, L38 (of the LaTeX source): typo ‘idenitifers’

Text S1: I’m a bit leery of the “individual reproduction number” terminology; the same wording has definitely been used for something else in the past (e.g. Fraser, PLOS ONE, 2007), and also I’m not really sure that the idea of a reproduction number attached to one person is coherent.

Table S2 does not compile in LaTeX.

Reviewer #2: After considering the other reviews, I understand that heuristic methods in tree (or forest of trees) comparison are of interest per se. I (Rev 2) interpreted the paper as essentially comparing outputs of statistical modelling of the same data using different inferential models and, in that perspective, I have had and still have doubts.

If a disease is human-to-human transmitted and one-to-one transmission is assumed, there is a "true" transmission tree. Given data on people involved in the transmission, one may try to "estimate" this tree.

Of course, the estimation may encounter various problems, such as undetected individuals, multiple introductions, imprecise infection times, etc..., but could also profit from data about location of individuals or other contact-tracing-like information. If, furthermore, MCMC-like methodology is used, the result will be an a posteriori distribution on the space of trees from which a sample (more or less iid) may be obtained. The distribution will depend on a priori assumptions made, in addition to data. This sample from the a posteriori is one way of obtaining a forest of trees.

If one considers two estimation procedures having different initial assumptions and/or data, the resulting a posteriori distributions should be different, as a rule. Whether this can be seen by comparing two respective samples from the a posteriori distributions (e.g by significance testing of the hypothesis that the two samples come from the same distribution) should then essentially depend on sample size, since one knows beforehand that the distributions are not the same...

In their answer to the previous review (Simulation design), the authors focus on different assumptions regarding offspring distributions defined by R0 and k. From a statistical point of view, one would then incourage the use of goodness-of-fit statistics and selection of best (or credibility set of best) offspring parameters, rather than significance testing between two alternatives...

I thus still think that, from a statistical methodological point of view, the paper does not have a clear aim, but, seen from a more empirical point of view, I have no special objection to the proposal and study of heuristic methods for comparing forests of trees ...

**Have the authors made all data and (if applicable) computational code underlying the findings in their manuscript fully available?**

Reviewer #1: Yes

Reviewer #2: None

PLOS authors have the option to publish the peer review history of their article (what does this mean?). If published, this will include your full peer review and any attached files.

Reviewer #1: No

Reviewer #2: No

**Figure resubmission:**
---

## [Decision Letter · Decision Letter 2]

27 Apr 2026

Dear Dr Geismar,

We are pleased to inform you that your manuscript 'A statistical framework for comparing epidemic forests' has been provisionally accepted for publication in PLOS Computational Biology.

Best regards,

Jennifer A. Flegg

Section Editor

PLOS Computational Biology

Jennifer Flegg

Section Editor

PLOS Computational Biology

Reviewer's Responses to Questions

**Comments to the Authors:**

Reviewer #2: Still reviewer 2... The paper is now clearer about what it does, although I still have doubts about the theoretical/formal underpinning of "comparing forests of trees". However, I think that the paper will be useful to many practitioners of "tree and forest creation" and therefore recommend its publication.

**Have the authors made all data and (if applicable) computational code underlying the findings in their manuscript fully available?**

Reviewer #2: None

PLOS authors have the option to publish the peer review history of their article (what does this mean?). If published, this will include your full peer review and any attached files.

Reviewer #2: No

---

## [Editor Report · Acceptance letter]

PCOMPBIOL-D-25-00809R2

A statistical framework for comparing epidemic forests

Dear Dr Geismar,

I am pleased to inform you that your manuscript has been formally accepted for publication in PLOS Computational Biology. Your manuscript is now with our production department and you will be notified of the publication date in due course.

With kind regards,

Zsofia Freund
